# Review of the Effectiveness of Various Adjuvant Therapies in Treating *Mycobacterium tuberculosis*

**Arman Amin** [1], **Artin Vartanian** [2], **Aram Yegiazaryan** [1] [ID], **Abdul Latif Al-Kassir** [1] **and Vishwanath Venketaraman** [1,*] [ID]

1   College of Osteopathic Medicine of the Pacific, Western University of Health Sciences, Pomona, CA 91766-1854, USA; arman.amin@westernu.edu (A.A.); aram.yegiazaryan@westernu.edu (A.Y.); abdullatif.alkassir@westernu.edu (A.L.A.-K.)
2   St. George's University School of Medicine, St. George's University, St. George's 999166, Grenada; avartani@sgu.edu
*   Correspondence: vvenketaraman@westernu.edu

**Abstract:** Tuberculosis disease is caused by the bacterium *Mycobacterium tuberculosis*. It is estimated that 10 million people have developed tuberculosis disease globally, leading to 1.4 million deaths in 2019. Treatment of tuberculosis has been especially challenging due to the rise of multidrug-resistant (MDR-TB) and extensive drug-resistant (XDR-TB) tuberculosis. In addition to drug-resistant genotypes, the standard treatment of tuberculosis by first-line agents is also challenging due to toxicity and costs. In the last four decades, there have only been two new anti-tuberculosis agents—bedaquiline and delamanid. Therefore, shorter, safer, and more cost-effective therapies are needed to adequately treat tuberculosis. In this review, we explore various adjuvants such as glutathione, everolimus, vitamin D, steroid, aspirin, statin, and metformin and their usefulness in reducing the burden of tuberculosis. Glutathione, everolimus, aspirin, and metformin showed the most promise in alleviating the burden of tuberculosis. Despite their potential, more clinical trials are needed to unequivocally establish the effectiveness of these adjuvants as future clinical therapies. Methods: The journals for this review were selected by conducting a search via PubMed, Google Scholar, and The Lancet. Our first search included keywords such as "tuberculosis" and "adjuvant therapy." From the search, we made a list of adjuvants associated with tuberculosis, and this helped guide us with our second online database search. Using the same three online databases, we searched "tuberculosis" and "respective therapy." The adjuvants included in the paper were selected based on the availability of sufficient research and support between the therapy and tuberculosis. Adjuvants with minimal research support were excluded. There were no specific search criteria regarding the timing of publication, with our citations ranging between 1979 to 2021.

**Keywords:** tuberculosis; adjuvant therapy; glutathione; everolimus; vitamin D; steroids; aspirin; statins; metformin

## 1. Introduction

Tuberculosis (TB) is an infectious disease caused by *Mycobacterium tuberculosis* (*Mtb*), a bacterium that first appeared roughly 20,000 years ago [1]. TB is transmitted from person to person via air droplets, primarily affecting the pulmonary system as tubercle bacilli infect lung alveoli. Bacilli may cause extrapulmonary TB, as they spread and infect other organ systems including the musculoskeletal system, lymphatic system, central nervous system, and abdominal organs. TB proves to be a significant burden on global public health, as the Global Tuberculosis Report 2020 declared it to be the leading cause of death from an infectious pathogen. The advancement to care and prevent TB has been slow and has not reached the WHO targets [2]. In addition, 10 million people developed TB around the globe, leading to a combined 1.4 million deaths among HIV-negative and -positive people in 2019 [2]. Today's widely accepted treatments for TB include a combination of first-line anti-TB agents: isoniazid (INH), rifampin (RIF), ethambutol (EMB), pyrazinamide (PZA), as

well as second-line treatments for TB such as amikacin and capreomycin [3]. Treatment of tuberculosis proves to be a challenge, as the spread of multidrug-resistant TB (MDR-TB) and extensively drug-resistant TB (XDR-TB) continue to rise [4]. Drug resistance is attributed to a variety of factors including disorganized treatment from structural weaknesses in health systems, community transmission, and facility-based transmission, as well as genetic and phenotypic mutations [4,5]. Treating TB, including MDR-TB and XDR-TB, is challenging due to a variety of factors including long duration of therapy, toxicity from first-line anti-TB agents, costs, and substandard outcomes. After four decades, there have been two new anti-TB agents to control MDR/XDR-TB—bedaquiline and delamanid [6]. It is imperative to explore new therapies that are affordable, limit antibiotic resistance, and have a low economic impact while limiting adverse effects. In this review, we will explore adjuvant therapies such as glutathione (GSH), everolimus, vitamin D, steroids, aspirin, statins, and metformin.

## 2. Adjuvants

### 2.1. Glutathione

Glutathione (GSH) is a tripeptide composed of γ-L-glutamyl-L-cysteinyl-glycine that is present in all mammalian tissues at concentrations of 1–10 mM, in the thiol-reduced (GSH) and disulfide-oxidized (GSSG) forms [7,8] (Meister 1988; Kaplowitz et al., 1985). There are three major reservoirs of GSH in eukaryotic cells. Nearly 85% of the cellular GSH is found in the cytosol, 10–15% is in the mitochondria, and a small amount is in the endoplasmic reticulum [9,10].

Its primary functions are involved in defending cells against oxidative stress and modulating DNA synthesis, apoptosis, immune function, detoxifying electrophiles, scavenging free radicals, and providing a cysteine reservoir [11,12].

The glutamate and cysteine of GSH are linked by a peptide bond between the γ-carboxyl group of glutamate rather than the α-carboxyl group, which makes it subject to hydrolysis only by γ-glutamyltranspeptidase (GGT), only present on the external surfaces of certain cell types. This makes GSH resistant to intracellular degradation because it is only metabolized by GGT extracellularly [11].

GSH is synthesized in the cytosol of all cells involving two ATP-requiring enzymatic steps: formation of γ-glutamylcysteine from glutamate and cysteine and then the formation of GSH from γ-glutamylcysteine and glycine. The first step of GSH synthesis is the rate-limiting step and is catalyzed by glutamate–cysteine ligase (GCL) composed of a heavy/catalytic subunit, GCLC, and a light/modifier subunit, GCLM [13]. The second step of GSH synthesis is catalyzed by GSH synthase, and it is subject to feedback inhibition by GSH [14].

GSH, in the presence of GSH peroxidase, functions to reduce hydrogen peroxide formed from oxidative stress during mitochondrial respiration, where no catalase is present. GSH is oxidized to GSSG, and GSSG reductase reduces it back to GSH, using NADPH, forming a redox cycle [15]. However, in the case of severe oxidative stress, the ability of the cell to reduce GSSG to GSH is diminished and leads to an accumulation of GSSG and depleted levels of GSH [16].

After intracellular infection by Mtb, the immune system attacks the bacteria by either killing them off or engulfing them in a specialized structure composed of immune cells known as granuloma. Granuloma is localized within the lungs and renders the TB inactive, which is referred to as latent TB [17]. The process of granuloma formation is mediated by various cytokines, such as tumor necrosis factor alpha (TNF-α), interleukin-2 (IL-2), IL-6, IL-12, and interferon gamma (IFN-y) [18]. Several studies have shown IL-2 to have effects on granuloma formation and suppressing the levels of viable Mtb; however, it has also been demonstrated that IL-2 plays an antagonistic role during combined chemotherapy and immunotherapy for Tb infections. TNF-a is an important factor in granuloma formation, as studies have shown that individuals with mutations in TNF-a promoters or patients treated with TNF antagonists show an increased risk of TB infections [19]. In

an individual who is immunocompromised, these granulomas may undergo liquefaction, releasing the bacteria and resulting in active TB. GSH levels are significantly compromised in individuals with active pulmonary TB [20]. Previous studies have exhibited that GSH has antimycobacterial and immune-modulating properties exhibited through their action on NK cells and by restoring TH1 cytokine response [21,22]. GSH, when combined with IL-2 and IL-12, enhances the functional activity of NK cells, making them able to inhibit the growth of Mtb in human cells [23]. IL-12 has also been shown to be effective as adjuvant therapy in TB individually. A case report of a 24-year-old male hospitalized in April 1998 in Germany, diagnosed with pulmonary TB with military spread and TB involvement of cervical lymph nodes, was given IL-12 therapy [24]. The adjuvant therapy was given for 3 months and the patient showed significantly improved clinical results. It was shown that the addition of IL-12 restored the impaired IFN-γ release, which helped further enhance T cell and macrophage activity [24]. As this was a single case, more research into IL-12 treatment should be evaluated, but this case should help encourage the study of IL-12 effects in TB patients.

It has been previously demonstrated that supplementation with liposomal GSH restored redox homeostasis, induced a cytokine balance, and improved immune responses against Mtb infection [22]. It was also shown in a further study, that macrophages treated with suboptimal levels of each of the first-line antibiotics in conjunction with N-acetyl cysteine (NAC) a precursor to GSH, resulted in a significant reduction in the intracellular survival of Mtb, compared to that of antibiotics alone. Combined treatment of NAC with antibiotics resulted in a decrease in the production of TNF-α, an inflammatory cytokine overexpressed in Mtb infection, and IL-10, an immunosuppressive cytokine, by the macrophages [25–27]. These findings suggest that GSH can be a potential adjunct treatment with the previously mentioned first-line antibiotics to clear Mtb infection.

### 2.2. Everolimus

Everolimus is an immunosuppressant drug that has been approved for use in organ transplant recipients and in the treatment of various forms of cancer [28,29]. As an analog to rapamycin, it similarly inhibits the mammalian target of the rapamycin (mTOR) pathway. Rapamycin is not typically used as a host-directed therapy for TB treatment as it has low bioavailability and a high half-life. Everolimus is a more ideal candidate for these purposes, as it is more soluble in water and has a lower half-life [30]. The mTOR pathway is an attractive target for host-directed therapies, as it is a key regulator of cellular growth and proliferation [31]. Previous research has indicated that inhibiting the mTOR pathway can lead to improved cellular survival and antibacterial defense by activating autophagy [32]. Autophagy is an essential response of the innate immune system as it removes intracellular debris introduced by foreign pathogens such as Mtb [33]. Studies have shown that autophagy is successful in eliminating Mtb [34]. As Everolimus inhibits the mTOR pathway, it leads to the induction of autophagy, which then augments cellular defense mechanisms against Mtb.

A 2014 study showed that everolimus modulated immune function in elderly volunteers after they received the 2012 influenza vaccine. Volunteers receiving everolimus displayed a significant decrease in the percentage of programmed cell death receptor-1 CD4 and CD8 T cells, which typically accumulate with age and provide diminished immune responses [35]. Another clinical trial showed that elderly volunteers receiving a combination of everolimus and dactolisib, both mTOR inhibitors, resulted in a decreased rate of infections and increased humoral responses to influenza vaccination [36]. Latent TB infections may lead to inflammatory cytokine responses resulting in the formation of lung granulomas. Everolimus treatment alone, as well as in conjunction with isoniazid was shown to significantly decrease levels of the pro-inflammatory cytokine TNF-a in a human granuloma model [37]. More recently, researchers assessed rifampin-susceptible pulmonary TB patients and their lung function by measuring FEV1 and found that patients treated with everolimus displayed enhanced FEV1 recovery at 180 days of treatment, compared

to the control group [38]. These studies demonstrate how everolimus can be beneficial as a host-directed adjuvant in the treatment of pulmonary TB.

While everolimus is efficient in stimulating the immune system, its use as an adjuvant in TB treatment may be counterproductive, as it has powerful immunosuppressive properties as well. The administration of everolimus has been associated with the activation of pulmonary TB. A case report from 2011 found that a kidney transplant recipient developed tuberculosis after 3 months of everolimus administration. The authors proposed that everolimus may have been given at too high of a dose, resulting in over suppression of the immune system and reactivation of latent TB [39]. In addition, researchers performing a matched case–control study, found that organ transplant recipients who received everolimus and hemodialysis were more likely to develop TB [40]. Everolimus must be carefully dosed and administered due to its immunosuppressive qualities. A study delivering rapamycin as an inhalable particle showed that it was more effective in the removal of intracellular Mtb when compared to rapamycin in solution [41]. Delivering everolimus as an inhalable particle to treat TB is worth exploring further, as it may prevent the unwanted consequences of over-immunosuppression. More studies should be conducted in order to better understand the effects of different Everolimus doses and delivery mechanisms with respect to the mTOR pathway and immunosuppression in the treatment of TB.

*2.3. Vitamin D*

As part of the adaptive immune response, different types of T cells exist to provide protection for the human body. T helper (Th) cells are one of the major types of T cells that release specific cytokines to mediate immune responses. Specifically, Th1 cells are critical for regulating intracellular infections from Mtb [42]. Vitamin D and 1,25 $(OH)_2D$ inhibit the proliferation of Th1 and Th17 cells and their pro-inflammatory cytokines such as IL-2, IFN-$\gamma$, IL-17 secretion, as well as induce T regulatory responses [42]. While the inhibition of Th1 and Th17 CD4 T cell responses would allow the pathogen to replicate and cause active disease, Vitamin D has a role in the innate immune response, which reduces the viability of Mtb (Figure 1). A 2006 paper showed that when human macrophages were activated by Toll-like receptors (TLR) there was an overexpression of vitamin D receptor (VDR) and the vitamin D-1-hydroxylase genes. This overexpression led to the destruction of Mtb via induction of the antimicrobial peptide cathelicidin [43]. This study showed that when human macrophages that had been infected with Mtb were treated with 1,25 $(OH)_2D_3$, there was a reduction in the number of viable bacilli [43]. While the innate immune system is responsible for the first-line defense, the adaptive immune response has a larger role in pathogen neutralization. Vitamin D is a primary target for antigen-presenting cells (APCs), a critical component of the adaptive immune system [44]. In addition, 1,25 $(OH)_2D_3$ regulates the cytokines and chemokines released by dendritic cells (DC) by inhibiting IL-12 and IL-23 cytokines from Th1 while upregulating IL-10 cytokine, which shows anti-inflammatory properties [44]. Due to Vitamin D's involvement in the inflammatory process, it is worth evaluating its efficacy in combating TB disease. In a vast majority of patients with TB, 25(OH)D levels fall below 20 ng/mL [45]. In 2018, a meta-analysis was conducted to investigate the association between vitamin D levels and children with TB showed that children with latent TB had significantly lower Vitamin D levels than controls [46]. A 2019 study performed to determine the impact of vitamin D levels and risk of TB disease showed vitamin D predicts TB disease in a dose-dependent manner [47]. Before antibiotic use, vitamin D was already used to treat TB as seen in an 1848 study that evaluated the role of sun exposure and administration of cod liver oil, both rich in Vitamin D [45]. In the study, 18% of patients who were treated with cod liver oil were stabilized, compared to 6% of the control group [48]. In 1998, a study showed that in children who took vitamin D supplementation along with first-line anti-TB drugs, clinical improvement occurred at a quicker rate [45]. Salahuddin et al. demonstrated that vitamin D supplemented in high doses leads to improvements in radiographs and weight gain in patients with TB. The study was limited in the fact that it did not show a difference

in sputum conversion rates, a clinical tool used to assess the efficacy of treatments [49]. However, there have also been studies that did not support clinical improvements. Table 1 summarizes the findings of key studies. A meta-analysis conducted in 2019 to analyze the effectiveness of vitamin D supplementation on pulmonary TB, indicated that vitamin D did not have any beneficial effect on anti-TB treatment. However, in participants with a tt genotype, sputum culture conversion times were shortened [50]. Based on the analyzed studies, vitamin D was most effective in shortening sputum conversion times but given the conflicting results, more randomized control trials are needed to establish its clinical efficacy. Therefore, the use of vitamin D supplementation is not unambiguously correlated to an enhanced anti-TB response.

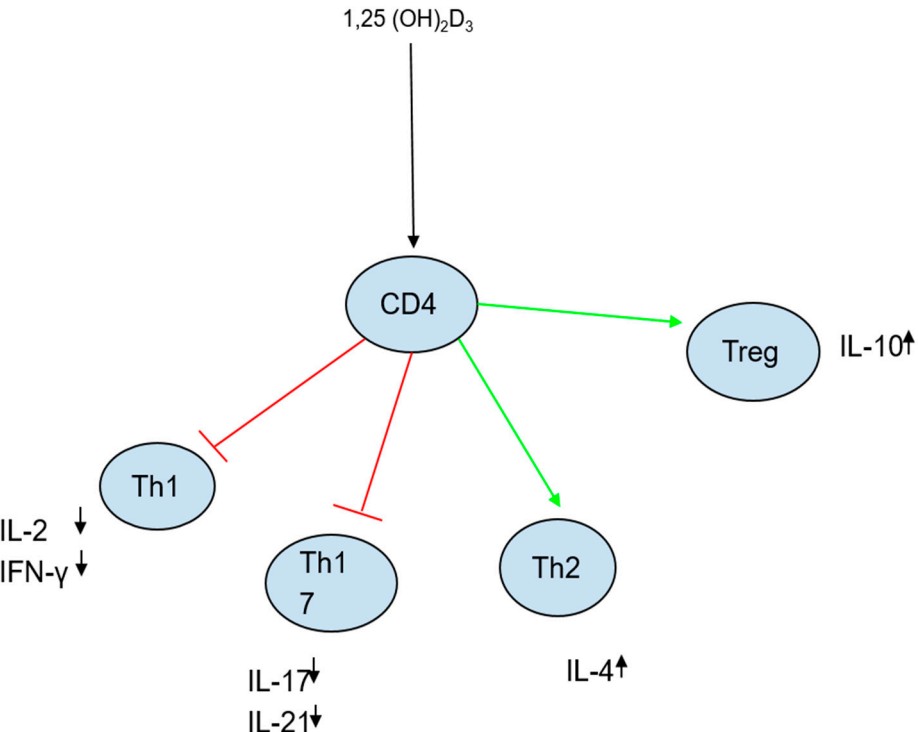

**Figure 1.** Immunomodulatory effects of 1,25 (OH)$_2$D.

**Table 1.** Overview of major clinical studies.

| Author | Population | Vitamin D Dose | Duration | Findings |
|---|---|---|---|---|
| Martineau et al. (Mathyssen, Carolien et al., 2017) | 146 adults | 2.5 mg at start, and at days 14, 28, and 42 | 56 days | No effect on sputum culture conversion on the overall population. |
| Wejse et al. (Mathyssen, Carolien et al., 2017) | 365 adults | 100,000 IU at start, 3 months, 5 months, and 8 months | 1 year | No effect on clinical outcome/mortality |
| Tukvadze et al. (Mathyssen, Carolien et al., 2017) | 199 adults | 50,000 IU 3×/week for 8 weeks, followed by every other week for another 8 weeks | 16 weeks | No improvement in sputum TB clearance |
| Ganmaa et al. (Ganmaa, Davaasambuu et al., 2020) [51] | 8851 children | 14,000 IU vitamin D$_3$ or placebo | 3 years | Not lower risk of TB infection |
| Sudfeld et al. (Sudfeld, Christopher R et al., 2020) [52] | 6250 HIV+ adults | 50,000 IU Vitamin D$_3$ for first month of ART followed by 2000 IU Vitamin D$_3$ daily | 3 years | No overall effect of supplementation on mortality risk. No difference in incidence of pulmonary TB between Vitamin D$_3$ vs. placebo |
| Morcos et al. [41] (Mathyssen, Carolien et al., 2017) | 24 children (<13 y/o) | 1000 IU daily. No placebo implemented | 8 weeks | Clinical improvement in radiography (X-ray and Ultrasound). Weight gain in patients |
| Nursyam et al. [41] (Mathyssen, Carolien et al., 2017) | 67 patients (15–59 y/o) | 0.25 mg daily × 6 weeks | 12 weeks | Increased rate of sputum conversion. Improved radiologic findings |
| Salahuddin et al. [41] (Mathyssen, Carolien et al., 2017) | 259 patients (>16 y/o) | 2 IM injections of 600,000 IU given at 1 month apart | 12 weeks | Faster clinical and radiographic improvement. Enhanced host immune activation |
| Hassanein et al. [41] (Mathyssen, Carolien et al., 2017) | 60 adults | 1 IM injection of 200,000 IU | 8 weeks | Enhanced TB score and more rapid sputum conversion rates |

*2.4. Steroids*

Cortisol is a glucocorticoid steroid hormone produced by the adrenal glands and is a key factor in a number of important physiological functions including glucose metabolism, stress reduction, metabolic function, immune system suppression, and anti-inflammatory processes. Due to powerful anti-inflammatory properties, synthetic steroids (Corticosteroids) such as dexamethasone and prednisone have been widely utilized in the treatment of many autoimmune and infectious diseases. As they have potent immunosuppressive capabilities and augment mucociliary clearance, corticosteroids are commonly used in treating pulmonary diseases including asthma, COPD, and pneumonia [53]. Yet, there are both recommendations for and against adjuvant corticosteroid use in the treatment of different forms of TB.

Although recent studies have shed light on the biological mechanisms of corticosteroid use in TB, the precise mechanism of action is poorly understood. Corticosteroids have been argued to have adverse effects in treating pulmonary TB, as long-term use can cause individuals susceptible to mycobacterial infection to develop TB, as well as reactivate latent bacilli within macrophages [54]. A 2017 study showed that glucocorticoids enhance mycobacterial survival by inhibiting TBK1 Kinase, which is a mediator of autophagosome maturation and nitric oxide production, and by downregulating genes promoting autophagy. Glucocorticoids also activate the Akt/mTOR pathway, which leads to autophagy inhibition [55]. These mechanisms demonstrate how glucocorticoids may be detrimental in TB treatment, as they suppress the immune system, facilitating mycobacterial survival and reactivation of latent TB. On the contrary, a group of investigators found that corticosteroids may be successful in attenuating a TB infection by protecting lung fibroblast cells from mycobacterial killing and reducing intracellular bacterial growth in human monocyte-derived macrophages [56]. Moreover, a study found that corticosteroids such as dexamethasone prevented necrotic cell death of Mtb infected cells by facilitating mitogen-activated protein kinase phosphatase 1 (MKP-1) dependent dephosphorylation of p38 MAPK. The researchers also identified several corticosteroids that protect TB infected cells as successfully as TB antibiotics [57]. These findings may explain how corticosteroid use can be beneficial in specific cases of TB treatment, as they protect cells from cytolysis and reduce the inflammatory effects of an active infection.

Regarding pulmonary TB, the use of corticosteroids is controversial as some researchers have argued it is beneficial, whereas others have shown it may worsen infection [58]. A study conducted in South Korea—an intermediate TB-burden country—showed that inhaled corticosteroid use was associated with an increase in developing TB [59]. A clinical trial analyzing the use of corticosteroids in HIV-associated TB found that prednisolone use was associated with increased Mtb clearing from sputum. However, it was also related to increased levels of HIV RNA transcription; therefore, the use of corticosteroids to treat TB in HIV-positive patients was not recommended [60]. Adjuvant corticosteroid treatment may be considered in HIV-positive patients if they are under sufficient antiretroviral therapy. Regarding advanced cases of pulmonary TB, a 2003 systematic review found that adjunctive use of corticosteroid therapy resulted in significant clinical and radiographic benefits in such patients [61]. The Cochrane Library provided an update to this review by assessing findings in 18 trials and found that adjunctive corticosteroids did not reduce mortality from pulmonary TB [62]. Given that corticosteroids may reactivate a latent TB infection and augment mycobacterial growth, in addition to a lack of strong evidence for reducing mortality, the use of corticosteroids as adjuvant therapy in pulmonary TB treatment must be carried out with caution. On the other hand, some studies indicate that adjuvant corticosteroid use may be beneficial across different types of extrapulmonary TB. As TB meningitis presents as one of the deadliest forms of extrapulmonary TB, it is critical to find effective adjuvant therapies that may decrease mortality. The World Health Organization strongly recommends dexamethasone or prednisolone tapered over 6–8 weeks in the initial treatment of Meningeal TB [63]. A systematic review analyzing nine trials found that adjuvant corticosteroid use reduced mortality in TB meningitis by

approximately one quarter [64]. Furthermore, dexamethasone treatment improved the survival rate in patients infected with TB who are at risk of death or disability in TB meningitis [65]. TB pericarditis is a complication of TB that is often missed and diagnosed late, resulting in cardiovascular issues and death in high TB burden countries [66]. The United Kingdom's National Institute for Health and Clinical Excellence recommends adjuvant corticosteroid uses in its guidelines for treating TB pericarditis, whereas the WHO makes a conditional recommendation [63,67]. A Cochrane Library systematic review found that corticosteroids likely reduced mortality in TB pericarditis patients who were HIV negative [68]. Lymph node TB (LNTB) is one of the more common forms of extrapulmonary TB and constitutes about 20–40% of such cases. A clinical trial conducted in 2016 found that 57 out of 60 patients receiving a gradually tapered dosage of adjuvant prednisolone for 4 weeks had complete resolution of LNTB, compared to 44 out of 60 patients in the group that did not receive the corticosteroid [69]. A recent trial also recommended the use of adjuvant prednisolone in treating LNTB as researchers observed higher rates of complete resolution and symptomatic improvement, as well as lower rates of complications in the patients that received prednisolone [70]. While there are studies showing benefits to using adjuvant corticosteroids in extrapulmonary TB, additional studies with larger sample sizes must be conducted in order to come up with stronger conclusions supporting their use.

### 2.5. Aspirin

Aspirin is an anti-inflammatory medication that acts by irreversibly inhibiting cyclooxygenase enzymes to decrease the production of prostaglandins and platelet aggregation [71]. As discussed, the current treatment for tuberculosis entails using antimycobacterial agents. However, aspirin targets host immune function and has been proposed as a possible adjuvant therapy for TB [72]. TB disease leads to the proliferation of neutrophils, leading to excess inflammation and exacerbation of tuberculosis. In a systematic review published in 2017 that evaluated clinical and animal studies, NSAIDs such as aspirin can be helpful when given as a co-therapy for TB [72]. NSAIDs specifically reduce the inflammatory response by inhibiting prostaglandin E2 (PGE2) [72]. PGE2 has been shown to promote apoptosis of macrophages that have been infected with TB [73]. Additionally, a study conducted on mice showed that aspirin in combination with pyrazinamide amplified the effects of pyrazinamide during the early phase of treatment [74]. This study was limited in that the long-term effects of NSAIDs in conjunction with pyrazinamide were not evaluated. In a case–control study, patients with type 2 diabetes mellitus and pulmonary TB were divided into two groups. In one group, aspirin was combined with anti-TB drugs, and the control group consisted of only anti-TB drugs. This study showed that the aspirin group had a significantly higher sputum negative conversion rate [75]. A retrospective study conducted by Misra et al. evaluated the benefit of using aspirins as an additional therapy to steroids and antituberculosis therapy for TB meningitis. Despite having a more severe form of TB meningitis, results showed that patients who received aspirin with steroids had fewer deaths [76]. Based on the analyzed studies, aspirin shows potential to be an effective adjuvant therapy when combined with first-line anti-TB agents.

### 2.6. Statin

Statins are a class of drugs that are used to lower cholesterol synthesis and are used in patients with hypercholesterolemia and coronary disorders [77]. These drugs are used to lower low-density lipoprotein (LDL) cholesterol levels to prevent the development of atherosclerosis [78]. Along with lowering cholesterol levels, statins have shown potential to be used as antimicrobial, immunomodulatory, and anti-inflammatory agents [79]. Statins promote autophagy, phagosome maturation, and decrease the growth of *M. tuberculosis* [80]. A 2014 study showed that 5 μM of simvastatin in combination with first-line anti-TB drugs significantly increased the tuberculocidal activity of isoniazid in vivo at 72 h post infection when compared to isoniazid alone [81]. However, what was observed in mice in this study may translate differently in humans due to the known drug–drug interactions of

simvastatin and isoniazid [82]. A case–control study conducted in Taiwan confirmed that long-term statin use has a protective effect against pulmonary TB [83]. Despite some promising results, more studies are needed on statins, specifically analyzing drug–drug interactions with first-line anti-TB agents, as well as on determining the appropriate statin doses.

### 2.7. Metformin

Metformin is an FDA-approved drug that is most widely used for the treatment of type II diabetes [84]. Diabetes has been shown to increase the susceptibility and severity of TB, but the mechanism of action is not fully understood at this time. It is thought that diabetic drugs, such as metformin, have several positive effects on humans due to their actions in glycemic control [85]. Thus, the possible use of metformin in combination with typical anti-TB medication has been of great interest to researchers with the hope of improving the effectiveness of TB treatment (Table 2).

**Table 2.** Overview of major clinical studies in regard to metformin.

| Author | Population | Metformin Dose | Duration | Findings |
|---|---|---|---|---|
| Lachmandas et al., 2019 | 11 healthy adults | Increasing dose starting at 500 mg to 1000 mg for 5 consecutive days. | 5 days | - Metformin can downregulate the Type 1 interferon pathway in humans.<br>- ROS production and phagocytosis activity were increased by metformin intake. |
| Marupuru et al., 2017 | Diabetics diagnosed with TB were the study group (SG = 152), and diabetics without TB were the control group (CG = 299). | 500 mg and 1000 mg | 8 months | - Metformin use is a protective agent against TB infection in diabetics and is not dose dependent.<br>- Poor glycemic control among diabetics is a risk factor for TB occurrence. |
| Nicholas R Degner et al., 2018 [86] | 2416 patients ≥ 13 years old undergoing TB treatment | Variable among patients | Retrospective cohort study between 2000 and 2013 | - Patients with DM had 1.91 times higher odds of death during TB treatment than patients without DM.<br>- Metformin use in patients with DM was significantly associated with decreased mortality during TB treatment. |

A crucial aspect of TB therapy is minimizing the growth of Mtb, which is the etiological agent of TB. Mtb growth success is dependent on the evasion of the bacilli from the host cell's innate and adaptive immune response [87]. Production of reactive oxygen species (ROS), reactive nitrogen species (RNS), and the destruction of intracellular pathogens using phagosomal machinery, otherwise known as the autophagy pathway, are significant features of a host cell's innate antimicrobial protection [88]. The use of the autophagy pathway is regulated by the mammalian target of rapamycin (mTOR) complex and adenosine monophosphate-activated protein kinase (AMPK), which act as activators to the pathway. Data from the 2014 research article by Singhal et al. suggest that metformin, which is an FDA approved-AMPK modulator and mTOR-independent autophagy activator, inhibits intracellular growth of Mtb and enhances the efficacy of conventional anti-TB drugs in mice models. The findings obtained from in vivo mice models infected with Mtb revealed that metformin selectively induced mitochondrial ROS (mROS) production, possibly due to metformin-mediated inhibition of NADH hydrogenase activity in the mitochondria, resulting in increased inhibition of intracellular growth of Mtb [88].

A 2017 case–control study was performed using diabetic human subjects with the objective of determining the protective effects of metformin against TB in individuals with diabetes [86]. The study's results showed that the patients who were on metformin had a protective effect against tuberculosis 3.9 times greater relative to the other hypoglycemic treatment regimens, and the use of metformin at any time was significantly associated with reduced risk of TB. In a more recent study conducted in 2019, the effects of metformin on host response to Mtb were investigated in healthy human subjects without diabetes

both in vivo and in vitro [85]. Their findings in regard to the in vitro study showed that metformin clearly affected cytokine production with significantly decreased Mtb lysate–induced production of TNF-α, IL-10, IFN-γ, and IL-17 by PBMCs and inhibited expression of IL18, IL23P19, and TGFB1 at the transcriptional level. Their findings also suggested that metformin decreased phosphorylation of p-p70S6K and p-4EBP1, which are downstream mTOR targets, while also increasing phosphorylation of AMPK, a known molecular mTOR target. They followed up the in vitro study with an in vivo study following 11 healthy subjects being given an increasing dose of metformin for 5 consecutive days (500 mg on days 1 and 2, increasing to 1000 mg on days 3–5). The results showed that the addition of metformin suppressed cytokine response to Mtb via inhibition of the Mtb–induced type 1 IFN response and inflammation in human PBMCs. Findings in these healthy human subjects also showed enhanced innate host defense pathways through strong upregulation in ROS production immediately after metformin treatment, as well as strong upregulation in several genes involved in the ROS production pathway [85]. A possible shortcoming of this study is the short duration (5 days) of the metformin exposure.

A scoping review conducted by Naicker, Sigal, and Naidoo discusses the repurposing of metformin as an Mtb therapy and the impact it has shown regarding mortality and morbidity of Mtb. They investigated ten studies in which metformin was used in association with front-line TB therapy, and two of the studies showed a significant reduction in mortality rate and morbidity in patients receiving both metformin and TB treatment [89]. Studies have also shown that metformin treatment is associated with a reduced incidence of latent TB, likely mediated by the enhanced Mtb-specific T cell immune response [90].

Overall, many studies have been conducted in regard to the adjuvant therapeutic capabilities of metformin in regard to TB treatment. The majority of studies have found that the use of metformin is effective in improving and enhancing host response to Mtb. Based on the results gathered from a large number of studies, metformin is a strong drug candidate to be used in conjunction with classic anti-TB medication to help better reduce the severity and mortality of TB.

### 3. Conclusions

From the conducted review in this study, everolimus, aspirin, and metformin show the most potential as effective adjuvants against TB. However, more clinical trials are needed to establish appropriate dosing and supplementation of these adjuvants with first-line anti-TB agents. While the current evidence for vitamin D, statins, and steroids are less conclusive, more studies are needed to further determine the efficacy of these agents. Steroids show promise against extrapulmonary TB, while vitamin D supplementation displayed conflicting results in regard to sputum conversion rates, and chronic statins use displayed a protective function against pulmonary TB. Although some of the research is not conclusive, further research into these adjuvants has the prospective capability to change how we treat TB. See Table 3 for a brief summary of the adjuvants discussed in this paper.

**Table 3.** Summary of adjuvant findings and conclusions.

| Adjuvant | Findings/Conclusions |
|---|---|
| Glutathione | These findings suggest that GSH can be a potential adjunct treatment with the previously mentioned first line antibiotics to clear Mtb infection via decrease in TNF-α and restoring redox homeostasis |
| Everolimus | More studies should be conducted |
| Vitamin D | Shorten sputum conversion times, but more randomized control trials are needed. |
| Steroids | Additional studies with larger sample sizes must be conducted in order to come up with stronger conclusions supporting their use. |
| Aspirin | aspirin shows potential to be an effective adjuvant therapy when combined with first-line anti-TB agents by reducing inflammation and amplifying the effects of anti-TB agents. |

**Table 3.** *Cont.*

| Adjuvant | Findings/Conclusions |
| --- | --- |
| Statins | more studies are needed with statins that lack drug–drug interactions with first-line anti-TB agents |
| Metformin | Metformin is a strong drug candidate to be used in conjunction with classic anti-TB medication to help better reduce the severity and mortality of TB through its proposed mechanism of inhibiting the mTOR and MAPK pathways. |

**Author Contributions:** A.A., A.V., A.Y. and A.L.A.-K. have contributed to drafting this review. V.V. conceived the framework, provided guidance and assistance, and made edits to the draft. All authors have read and agreed to the published version of the manuscript.

**Funding:** We appreciate the funding support from National Institutes of Health (NIH) award RHL143545-01A1.

**Institutional Review Board Statement:** Not applicable.

**Informed Consent Statement:** Not applicable.

**Data Availability Statement:** Data sharing not applicable. No new data were created or analyzed in this study. Data sharing is not applicable to this article.

**Acknowledgments:** We appreciate the funding support from NIH (RHL143545-01A1).

**Conflicts of Interest:** The authors declare no conflict of interest.

**Abbreviations List**

| | |
| --- | --- |
| Multidrug resistant | (MDR) |
| Extensive drug resistant | (XDR) |
| Isoniazid | (INH) |
| Rifampin | (RIF) |
| Ethambutol | (EMB) |
| Pyrazinamide | (PZA) |
| Multidrug resistant TB | (MDR-TB) |
| Extensively drug-resistant TB | (XDR-TB) |
| Tuberculosis | (TB) |
| Mycobacterium tuberculosis | (Mtb) |
| Glutathione | (GSH) |
| Reduced glutathione | (GSH) |
| Oxidized glutathione | (GSSG) |
| Γ-glutamyltranspeptidase | (GGT) |
| Glutamate–cysteine ligase | (GCL) |
| Tumor necrosis factor alpha | (TNF-$\alpha$) |
| Interferon gamma | (IFN-y) |
| Mammalian target of rapamycin | (mTOR) |
| T helper | (Th) |
| Toll-like receptors | (TLR) |
| Vitamin D receptors | (VDR) |
| Antigen-presenting cells | (APC) |
| Dendritic cells | (DC) |
| Mitogen-activated protein kinase phosphatase | (MKP-1) |
| Lymph node TB | (LNTB) |
| Prostaglandin E2 | (PGE2) |
| Low-density lipoprotein | (LDL) |
| Reactive oxygen species | (ROS) |
| Reactive nitrogen species | (RNS) |
| Adenosine monophosphate-activated protein kinase | (AMPK) |
| Mitochondrial ROS | (mROS) |

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
