# Peer review of "Review of the Effectiveness of Various Adjuvant Therapies in Treating Mycobacterium tuberculosis"

_2036-7449, doi:10.3390/idr13030074_

Round 1

Reviewer 1 Report

The review seems well-prepared and an important summary. However, there might be some papers that are missing and might be worth to add (papers and citations therein):

- I suggest to check papers and citations therein [please answer paper-by-paper]:

https://pubmed.ncbi.nlm.nih.gov/30259757/

https://jitc.biomedcentral.com/articles/10.1186/s40425-019-0717-7

https://www.neurologyindia.com/article.asp?issn=0028-3886;year=2018;volume=66;issue=6;spage=1678;epage=1679;aulast=Yadav

https://www.ncbi.nlm.nih.gov/pmc/articles/PMC4484774/;  https://doi.org/10.2217/fmb-2018-0110;  https://academic.oup.com/cid/article/41/2/201/530346;  https://erj.ersjournals.com/content/17/5/1049;  https://www.nature.com/articles/s41385-019-0226-5;  https://www.frontiersin.org/articles/10.3389/fmicb.2020.00435/full;  https://njms.rutgers.edu/departments/medicine/infectious_diseases/documents/CIDReconsidering.pdf;  https://pubmed.ncbi.nlm.nih.gov/26073420/

- It might be advantageous to include a summarizing table to compare the adjuvant therapies - mainly adding the conclusions of the single subchapters
- the numbering of the subchapters should be checked, probably there should be a "2. Adjuvants" followed by "2.1 Glutathione" etc.
-  Immunotherapy is not mentioned, but it is not clear, why.

Author Response

REVIEWER #1

We truly appreciate your comprehensive and thorough review of our paper. We have taken each comment to heart and to the best of our ability, made revisions to improve upon our original paper. We thank you for the feedback.

REVIEW COMMENTS

check papers and citations therein [please answer paper-by-paper]:

https://pubmed.ncbi.nlm.nih.gov/30259757/

  • We found the papers very interesting, however, it is discussing the use of efflux inhibitors, one of which is 2-aminothiazole UPAR-174, and we believe that it doesn’t correlate well with any of the adjuvant therapy sections we currently have. We did not think these three resources alone would be sufficient enough to create a dedicated section for it in our paper. However, if you think adding a new section on the novel use of efflux inhibitors would be beneficial then we will do further research into the topic.
  • https://www.ncbi.nlm.nih.gov/pmc/articles/PMC4484774/
  • https://doi.org/10.2217/fmb-2018-0110

https://jitc.biomedcentral.com/articles/10.1186/s40425-019-0717-7

  • This paper discusses a PD1 inhibitor for the use of tuberculosis, but no clear conclusion can be drawn. It states “In general, in case of active tuberculosis, ICPIs are temporarily withheld, any further immunosuppression is discontinued, and anti-tuberculosis treatment is timely initiated. Also, in patients diagnosed with either active or latent tuberculosis, it is not clear how long after the corresponding anti-TB treatment ICPIs should be safely resumed or initiated, with a duration of 2–4 weeks to be suggested. The continuously expanded implementation of IPCIs in cancer treatment requires resolving of these challenges by upcoming research data in order to maximize the clinical benefits of immunotherapy uninterruptedly and safely.”
  • Furthermore, upon inquiring more about PD-1 inhibitors we found that the cost of a PD-1 inhibitor, Keytruda was $5,000 for 4mL( https://www.drugs.com/price-guide/keytruda). We wanted to focus on more available, accessible, and affordable adjuvants as mentioned in the introduction for TB therapy.

https://www.neurologyindia.com/article.asp?issn=0028-3886;year=2018;volume=66;issue=6;spage=1678;epage=1679;aulast=Yadav

  • “A retrospective study conducted by Misra et al evaluated the benefit of using aspirins as an additional therapy to steroids and antituberculosis therapy for TB meningitis. Despite having a more severe form of TB meningitis, results showed that patients who received aspirin with steroids had fewer deaths (76).”
  • Yadav, Ravi. “Role of aspirin as an adjuvant therapy in tuberculous meningitis in adults: The time has come for a phase III randomized controlled trial.” Neurology India vol. 66,6 (2018): 1678-1679. doi:10.4103/0028-3886.246225

https://academic.oup.com/cid/article/41/2/201/530346

  • This paper discusses the therapeutic implications of IL-2 and IFN-gamma cytokines in response to tuberculosis. We feel that we have adequately addressed various cytokines and their response to tuberculosis. To be specific, we discussed the cytokines in sections 2.1 and 2.7. However, if you feel that we should discuss cytokine therapy as its own individual section then we will be happy to do so.

https://erj.ersjournals.com/content/17/5/1049

  • “IL-12 has also both been shown to be effective as an adjuvant therapy in TB individually. A case report of a 24 year old male hospitalized in April 1998 in Germany, diagnosed with pulmonary TB with military spread and TB involvement of cervical lymph nodes, was given IL-12 therapy (24). The adjuvant therapy was given for 3 months and the patient showed significantly improved clinical results. It was shown that the addition of IL-12 restored the impaired IFN-γ release which helped further enhance T cell and macrophage activity (24). As this was a single case, more research into IL-12 treatment should be evaluated for, but this case should help encourage the study of IL-12 effects in TB patients.”
  • Greinert U, Ernst M, Schlaak M, Entzian P. Interleukin-12 as successful adjuvant in tuberculosis treatment. Eur Respir J. 2001 May;17(5):1049-51. doi: 10.1183/09031936.01.17510490. PMID: 11488308.

https://www.nature.com/articles/s41385-019-0226-5

  • “Furthermore, dexamethasone treatment improved the survival rate in patients infected with TB who are at risk of death or disability in TB meningitis (65).”
  • Young, C., Walzl, G. & Du Plessis, N. Therapeutic host-directed strategies to improve outcome in tuberculosis. Mucosal Immunol 13, 190–204 (2020). https://doi.org/10.1038/s41385-019-0226-5

https://www.frontiersin.org/articles/10.3389/fmicb.2020.00435/full

  • We utilized this resource to discuss the impact Metformin has on morbidity and mortality in regards to Tb.
  • “A scoping review conducted by Naicker, Sigal, and Naidoo discusses the repurposing of Metformin as a Mtb therapy, and the impact it has shown regarding mortality and morbidity of Mtb. They investigated ten studies where Metformin was used in association with front-line TB therapy, and two of the studies showed a significant reduction in mortality rate and morbidity in patients receiving both Metformin and TB treatment (89).”
  • Naicker, N., Sigal, A., & Naidoo, K. (2020). Metformin as Host-Directed Therapy for TB Treatment: Scoping Review. Frontiers in Microbiology, 11. https://doi.org/10.3389/fmicb.2020.00435

https://njms.rutgers.edu/departments/medicine/infectious_diseases/documents/CIDReconsidering.pdf

  • “Several studies have shown IL-2 to have effects on granuloma formation and suppressing the levels of viable Mtb, however, it has also been demonstrated that IL-2 plays an antagonistic role during combined chemotherapy and immunotherapy for Tb infections. TNF-a is an important factor in granuloma formation as studies have shown that individuals with mutations in TNF-a promoters, or patients treated with TNF antagonists show increased risk of TB infections (19).”
  • Wallis, R. (2005). Reconsidering Adjuvant Immunotherapy for Tuberculosis. Clinical Infectious Diseases, 41(2), 201-208. doi: 10.1086/430914

https://pubmed.ncbi.nlm.nih.gov/26073420/

  • Used this source to show another way in which the use of Metformin can positively impact TB treatment/prevention.
  • “Studies have also shown that Metformin treatment is associated with reduced incidence of latent TB, likely mediated by the enhanced Mtb-specific T cell immune response (90).”
  • Rayasam, Geetha Vani, and Tanjore S Balganesh. “Exploring the potential of adjunct therapy in tuberculosis.” Trends in pharmacological sciences vol. 36,8 (2015): 506-13. doi:10.1016/j.tips.2015.05.005

- It might be advantageous to include a summarizing table to compare the adjuvant therapies - mainly adding the conclusions of the single subchapters

Adjuvant

Findings/conclusion

Glutathione

These findings suggest that GSH can be a potential adjunct treatment with the previously mentioned first line antibiotics to clear Mtb infection via decrease in TNF-α and restoring redox homeostasis

Everolimus

More studies should be conducted

Vitamin D

Shorten sputum conversion times, but more randomized control trials are needed.

Steroids

Additional studies with larger sample sizes must be conducted in order to come up with stronger conclusions supporting their use.

Aspirin

aspirin shows potential to be an effective adjuvant therapy when combined with first line anti-TB agents by reducing inflammation and amplifying the effects of anti-TB agents.

Statins

more studies are needed with statins that lack drug-drug interactions with first line anti-TB agents

Metformin

Metformin is a strong drug candidate to be used in conjunction with classic anti-TB medication to help better reduce severity and mortality of TB through its proposed mechanism of inhibiting the mTOR and MAPK pathways.

- the numbering of the subchapters should be checked, probably there should be a "2. Adjuvants" followed by "2.1 Glutathione" etc.

  • Added an adjuvant section as advised with the appropriate numbering of subsections.

Immunotherapy is not mentioned, but it is not clear, why.

  • Introduction mentions evaluating therapies that are affordable and economic. Immunotherapy access is limited in middle and low income countries. We looked into the suggestion, but felt that it didn’t fit into the paper as well as other therapies.

Reviewer 2 Report

This review is a comprehensive survey of major scientific databases for information on adjuvant therapies in treating tuberculosis. The work is interesting and well-written. I only have some minor observations that are listed in the following lines.

  1. How were the references for this review selected?
  2. Line 21: the introduction should be section 1. Please fix the numbering throughout the manuscript.
  3. Line 30: Avoid starting the sentence with a number that is not written out.
  4. Line 34: Give examples of the second line of treatments.
  5. Lines 42-44: Rephrase.
  6. Line 99: Add a reference.
  7. Lines 170-171: Give more details.
  8. Table 1: Use the same citation style for references.
  9. Table 1: Most of the studies mentioned in table 1 are using the same reference. Is this correct?
  10. Add a list of abbreviations.
  11. Add the author contributions and conflict of interest.

Author Response

REVIEWER#2

We truly appreciate your comprehensive and thorough review of our paper. We have taken each comment to heart and to the best of our ability, made revisions to improve upon our original paper. We thank you for the feedback.

Comments and Suggestions for Authors

This review is a comprehensive survey of major scientific databases for information on adjuvant therapies in treating tuberculosis. The work is interesting and well-written. I only have some minor observations that are listed in the following lines.

  1. How were the references for this review selected?
    1. Apologies for not including this in the original paper. The methods section has been added as follows.
    2. “Methods: The journals for this review were selected by conducting a search via PubMed, google scholar, and The Lancet. Our first search included keywords like “Tuberculosis” AND “adjuvant therapy”. From the search, we made a list of adjuvants associated with tuberculosis, and this helped guide us with our second online database search. Using the same three online databases, we searched “Tuberculosis” AND the respective therapy. The adjuvants included in the paper were selected based on the availability of sufficient research and support between the therapy and tuberculosis. Adjuvants with minimal research support were excluded. There were no specific search criteria regarding the timing of publication, with our citations ranging between 1979 to 2021.”

  1. Line 21: the introduction should be section 1. Please fix the numbering throughout the manuscript.
    1. Numbering edited in paper and added adjuvant section with subchapters per other reviewers suggestion.
  2. Line 30: Avoid starting the sentence with a number that is not written out.
    1. Sentence has been edited as follows with bolded terms showing changes.
    2. Ten million people developed TB around the globe, leading to a combined 1.4 million deaths among HIV negative.
  3. Line 34: Give examples of the second line of treatments.
    1. Sentence has been edited as follows with bolded terms showing changes.
    2. “second line treatments for TB such as amikacin and capreomycin”
  4. Lines 42-44: Rephrase.
    1. Sentence has been edited as follows with bolded terms showing the changes.
    2.  It is imperative to explore new therapies that are affordable, limit antibiotic resistance, have a low economic impact, while limiting adverse effects.
  5. Line 99: Add a reference.
    1. Reference has been added: “Everolimus is an immunosuppressant drug that has been approved for use in organ transplant recipients and in the treatment of various forms of cancer (28, 29).”
    2. Hasskarl J. Everolimus. Recent Results Cancer Res. 2018;211:101-123. doi: 10.1007/978-3-319-91442-8_8. PMID: 30069763.
    3. Guan TW, Lin YJ, Ou MY, Chen KB. Efficacy and safety of everolimus treatment on liver transplant recipients: A meta-analysis.     Eur J Clin Invest. 2019 Dec;49(12):e13179. doi: 10.1111/eci.13179. Epub 2019 Nov 19. PMID: 31610022

  1. Lines 170-171: Give more details.
    1. Sentence has been edited as follows with bolded terms showing changes.
    2. Before antibiotic use, vitamin D was already used to treat TB as seen by an 1848 study that evaluated the role of sun exposure and cod liver oil, both rich in Vitamin D, in TB (41). In the study, 18% of patients who were treated with cod liver oil were stabilized compared to 6% of the control group (42).
  2. Table 1: Use the same citation style for references.
    1. ​​Citations have been adjusted to have the same style for each.
    2. Sudfeld et al (Sudfeld, Christopher R et al 2020)(47).
  3. Table 1: Most of the studies mentioned in table 1 are using the same reference. Is this correct?
    1. Yes, most of the studies were from the same reference, but it was a review paper that highlighted different research studies. The table is also citing other papers with different research studies (i.e. Mathyssen, Carolien et al).

  1. Add a list of abbreviations.

Abbreviations List:

multidrug resistant (MDR)

extensive drug resistant (XDR)

isoniazid (INH)

rifampin (RIF)

ethambutol (EMB)

pyrazinamide (PZA)

multidrug resistant TB (MDR-TB)

extensively drug resistant TB (XDR-TB)

Tuberculosis (TB)

Mycobacterium tuberculosis (Mtb)

glutathione (GSH)

reduced glutathione (GSH)

oxidized glutathione (GSSG)

γ-glutamyltranspeptidase (GGT)

glutamate cysteine ligase (GCL)

tumor necrosis factor alpha (TNF-α)

interferon gamma (IFN-y)

mammalian target of rapamycin (mTOR)

T helper (Th)

Toll-like receptors (TLR)

Vitamin D receptors (VDR)

Antigen presenting cells (APC)

Dendritic cells (DC)

Mitogen-activated protein kinase phosphatase (MKP-1)

Lymph node TB (LNTB)

Prostaglandin E2 (PGE2)

Low density lipoprotein (LDL)

Reactive oxygen species (ROS)

Reactive nitrogen species (RNS)

Adenosine monophosphate-activated protein kinase (AMPK)

Mitochondrial ROS (mROS)

  1. Add the author contributions and conflict of interest.
    1. Once again, apologies for not including this in the original paper. The proper edits have been made and included as follows:
    2. Author Contributions
      1. “A.A. A.V. A.Y. A.L.A have contributed to drafting this review. V.V. conceived the framework, provided guidance and assistance, and made edits to the draft. The authors declare no conflicts of interest.”

  1. line 87 (Venketaraman’s laboratory previously demonstrated that…) the Editor in Chief suggests to rephrase the wording, avoiding self quotation.
  2. Sentence has been edited as follows with bolded terms showing changes.
  3. It has been previously demonstrated that supplementation with liposomal GSH restored redox homeostasis, induced a cytokine balance, and improved immune responses against Mtb infection (21).